# LEARNING INVARIANT FEATURES FOR ONLINE CONTINUAL LEARNING

## ABSTRACT

It has been shown recently that learning only *discriminative features* sufficient to separate the classes has a major shortcoming for continual learning (CL). This is because many features that are not learned may be necessary for distinguishing classes of some future tasks. When such a future task arrives, these features have to be learned by updating the network, which causes *catastrophic forgetting* (CF). A recent online CL work showed that if the learning method can learn as many features as possible from each class, called *holistic representations*, CF can be significantly reduced to achieve a large performance gain. This paper argues that learning only holistic representations is still insufficient. The learned representations should also be *invariant* and those features that are present in the data but are irrelevant to the class (e.g., background information) should be ignored for better generalization across tasks. This new condition further boosts the performance significantly. This paper proposes several strategies and a loss to learn *holistic and invariant representations* and evaluates their effectiveness in online CL. [1]

## 1 INTRODUCTION

A major challenge of continual learning (CL) is *catastrophic forgetting* (CF) (McCloskey & Cohen, 1989), which is caused by updating network parameters learned from previous tasks in learning a new task. Although many empirical approaches have been proposed to deal with CF, limited theoretical work has been done to study the necessary conditions for CL to overcome CF. Recently, Guo et al. (2022) argued that it is necessary to learn *holistic representations* of the data. This work proposes another condition, *invariance*, and argues that the learning of each task itself needs to be improved so that future tasks would not need to make major changes to old parameters.

It is well-known that supervised learning losses (e.g., cross-entropy) learn only discriminative features that are sufficient to separate the classes in a task. This is problematic for CL due to two main reasons. (1) many features that are not learned may be necessary to distinguish classes of some future tasks. When such a future task comes, these features have to be learned, which may make significant changes to the existing parameters and cause CF. (2) even if the previous parameters are completely protected, the classes in the new task still make the classification challenging because each task only learns discriminative features for its own classes, which causes *confusion* in classification when we need to classify all classes learned so far. We call this ***biased representation learning***. For example, task-1 learns to classify *black pig* and *dove*. The learner may only learn the color features (e.g., black and white) as they are sufficient to classify the two classes. However, task-2 learns rabbit and cow, which can be black or white. The color features learned from task-1 are no longer sufficient. Shape-based features need to be learned, which can make major changes to the existing parameters and cause CF. If the shape-based features had been learned from task-1, learning of task 2 will not need to update the parameters linked to the representation of pigs and doves as much, which gives less CF (mitigating (1) above). Since there are now 4 classes to classify, it can be confusing. Some cows or rabbits may be classified as pigs or doves due to the same color. Some pigs may be classified as cows or rabbits due to shape-based features learned in task-2. However, if all features have been learned in task-1 and task-2, such wrong classifications will be reduced (dealing with (2)). Recently, Guo et al. (2022) proposed to learn *holistic representations* from the input data to cover as many characteristics of the input as possible. Their system OCM

---

[1]The code has been submitted in the supplemental material.

learns holistic representations in online CL by maximizing the mutual information (MI) between the input data and the learned feature representations to ensure that as much information in the input is reflected in the learned features. This results in a major performance gain in online CL.

This paper argues that learning *holistic representations* is still sub-optimal. It is also necessary for CL to learn features that are *invariant* for each class. Those features that are present in the input but are irrelevant to the class should be ignored for better generalization to future tasks. For example, to classify images of *apple* and *fish*, some green background and red color of apple may be learned, but these features are not invariant to apple. When new classes *cows* and *ladybird* need to be learned, the feature green (or red) is shared (see Figure 1(a)) and may cause high logit outputs to apple and cow (or ladybird). Then the learner has to modify the representations of apple to reduce its logit values, which causes CF. That is, variant features are unsuitable for establishing decision boundaries. If the learner has learned shape and other invariant features for apple, the input of cow (or ladybird) will not activate the parameters linked to the apple representation. Then, in learning cow and ladybird, changes to the parameters that are important for apple will be limited, resulting in less CF. This paper aims to learn *invariant* and *holistic* representations for each class. This additional *invariance* condition gives another boost to online CL performance. Note that invariance is not critical for traditional supervised learning due to the i.i.d assumption,[2] but for CL, it is very important because each new task introduces new distributions and we want each class to be distinguishable against any past and future classes, which may have similar variant features that can confuse the model.

This paper works in the online *class-incremental learning* (CIL) setting.[3] It proposes a replay-based method called IFO (*I*nvariant *F*eature learning for *O*nline CL), which adds the invariance condition to holistic representation learning to also learn invariant features. We propose ***two new methods*** and ***one new optimization objective*** to achieve invariance. The first method is to construct a diverse set of environments and force the model to learn invariant features across the environments. The second method is a novel use of the replay data to learn invariant features and to deal with a ***local sampling bias*** issue. Finally, we combine the methods and propose a new optimization goal to learn invariant features. Theoretical justifications are also given. We verify the effectiveness of IFO in three online CL scenarios: traditional *disjoint task scenario*, *blurry task boundary scenario* and *data shift scenario*. The results show the proposed IFO outperforms strong baselines by a large margin.

## 2 RELATED WORK

Although many CL approaches have been proposed, little work has been done to study the necessary conditions for CL. The *replay* approach saves a small amount of past data and uses it to protect/adjust the previous knowledge in learning a new task (Rebuffi et al., 2017; Wu et al., 2019; Hou et al., 2019; Chaudhry et al., 2020; Zhao et al., 2021; Korycki & Krawczyk, 2021; Sokar et al., 2021; Yan et al., 2021; Wang et al., 2022a). *Pseudo-replay* generates replay samples (Shin et al., 2017; Hu et al., 2019; Sokar et al., 2021). Using *regularizations* to penalize changes to important parameters of previous tasks is another approach (Kirkpatrick et al., 2017; Ritter et al., 2018; Ahn et al., 2019; Yu et al., 2020; Zhang et al., 2020). *Parameter-isolation* approaches protect models of old tasks using masks and/or network expansion (Ostapenko et al., 2019; von Oswald et al., 2020; Li et al., 2019; Hung et al., 2019; Rajasegaran et al., 2020; Abati et al., 2020; Wortsman et al., 2020; Saha et al., 2021). Zhu et al. (2021) found that using data augmentations can learn more transferable features.

Online CL methods are mainly based on replay. ER randomly samples the replay data (Chaudhry et al., 2020), MIR chooses replay samples whose losses increase most (Aljundi et al., 2019a), ASER uses the Shapley value theory (Shim et al., 2021), and GDumb produces class balanced replay data (Prabhu et al., 2020). GSS diversifies the gradients of the replay data (Aljundi et al., 2019b). DER++ uses knowledge distillation (Buzzega et al., 2020), SCR uses contrastive loss (Mai et al., 2021), and NCCL calibrates the network (Yin et al., 2021). Applications of online CL are also reported (Yan et al., 2021; Wang et al., 2021). Bang et al. (2021) and Bang et al. (2022) proposed two blurry online CL settings. IFO is also a replay method but focuses on learning invariant features.

Domain generalization (DG) is also related. DG learns a model with inputs from multiple given source domains with the same class labels and test with inputs from unseen target domains. Ex-

---

[2]When out-of-distribution data or data shift is involved, invariance is also important (Arjovsky et al., 2019).

[3]In the CIL setting, no task related information (e.g., task-id) is provided in testing. The other popular CL setting is *task-incremental learning* (TIL), which needs the task-id to be provided for each test instance.

isting DG methods typically leverage data augmentations to expand the diversity of the source domains (Wang et al., 2020; Wu et al., 2020; Arjovsky et al., 2019; Rame et al., 2022). There are also works on a single domain. Yang et al. (2021) used a pre-trained domain augmenter to create novel stylized images. Yue et al. (2019) used auxiliary datasets to help create images. Our training data have no identified domains and we use no additional datasets. DG does not do CL.

## 3 PROBLEM FORMULATION

This paper uses three online CL settings. (1) traditional **disjoint tasks setting**, where the system learns a set of tasks incrementally one after another. Each task consists of several classes. The data for each task comes in a stream and the learner sees the data only once. In a replay method, when a small batch of data of the current task $t$ arrives from the data stream $D_t^{new} = (X_t^{new}, Y_t^{new})$ (where $X_t^{new}$ is a set of new samples and $Y_t^{new}$ is its set of corresponding labels), a small batch of replay data $D_t^{buf} = (X_t^{buf}, Y_t^{buf})$ is sampled from the memory buffer $\mathcal{M}$ and used to jointly train in one iteration. (2) **blurry task setting** (Koh et al., 2021), where the classes from a previous task may appear again later (more details in Sec. 5.2). (3) **data environment shift setting** (see details in Sec. 5.3).

Our model $F$ consists of two parts: the feature extractor $f_\theta$ and the classifier $\sigma_\phi$. $f_\theta$ extracts features from the input $x$ to form a high-level representation and $\sigma_\phi$ and the softmax operation map the representation to a prediction probability for each class.

**Learning holistic representations.** As discussed in Sec. 1, $f_\theta$ should learn *holistic feature representations* (Guo et al., 2022) but cross-entropy loss $\mathcal{L}_{ce}$ alone is unable to do that. Geirhos et al. (2020) also found that $\mathcal{L}_{ce}$ encourages models to stop learning once simple features suffice to minimize the loss. This means $f_\theta$ may learn simple discriminative features for the current distribution and ignore other semantic features present in the input data. OCM (Guo et al., 2022) tackles this problem by learning as many features as possible from the input $X$ via maximizing the mutual information between the input $X$ and the hidden representation $f_\theta(X)$ ($I(X; f_\theta(X))$) and the mutual information ($I(.)$) between the representation $f_\theta(X)$ and the label $Y$ ($I(Y; f_\theta(X))$) to learn as many characteristics of the input as possible. OCM uses the InfoNCE loss (Poole et al., 2019) ($\mathcal{L}_{InfoNCE}$) to optimize $I(X; f_\theta(X))$ (i.e., $I(X_t^{new}; f_\theta(X_t^{new})) + I(X_t^{buf}; f_\theta(X_t^{buf}))$) for the replay-based method. Note that it maximizes $I(Y; f_\theta(X))$ by using only $X_t^{buf}$ to calculate the cross-entropy loss as $I(Y; f_\theta(X))$ is maximized when $Y$ follows the uniform distribution. $X_t^{buf}$ is sampled from the class uniformly distributed data of all seen classes stored in the memory buffer. The loss for OCM is

$$\mathcal{L}_{OCM}(D_t^{buf}, D_t^{new}) = \mathcal{L}_{InfoNCE}(X_t^{new}, f_\theta(X_t^{new})) + \mathcal{L}_{InfoNCE}(X_t^{buf}, f_\theta(X_t^{buf})) + \mathcal{L}_{InfoNCE}(f_\theta(X_t^{buf}), f_\theta^{prev}(X_t^{buf}))$$
(1)

where the last term tries to prevent $F$ from forgetting features of previous classes by maximizing the mutual information between the hidden representations of $X_t^{buf}$ given by the current feature extractor $f_\theta$ and the feature extractor $f_\theta^{prev}$ before the current task, i.e., $I(f_\theta(X_t^{buf}); f_\theta^{prev}(X_t^{buf}))$.

**Invariant features.** As discussed in Sec. 1, in addition to learning holistic feature representations of the input data, this paper argues that the learned feature should also be invariant to each class. We give a more rigorous definition of the invariant class-related feature here.

**Definition 1** A feature $s$ is *invariant* for a class label $y \in Y$ if the following holds,

$$\mathbb{E}_{x \sim X_z}[\mathbb{I}(s \in x) \cdot |Corr(x, y)|] > \rho \quad \forall z \in Z$$
(2)

where $X_z$ is the input variable under the environment $z$, $Z$ is the set of all possible environments and not limited to the dataset $D$, $Corr$ is a function computing the correlation between the input $x$ and its label $y$, $| \cdot |$ is the absolute value function and $\rho$ is a constant. $\mathbb{I}(s \in x)$ is the indicator function that identifies if feature $s$ exists in input $x$. Note that by environment $z$ we not only mean the background of an object (e.g., forest as the background) but also the object in the input (e.g., a building showing different colors in rainy days and sunny days).

**Our goal.** Combining the two concepts, we aims to learn *holistic invariant representations* for each class, i.e., learning invariant features as many as possible, which help reduce CF and improve the generalization power. Below, we focus on proposing techniques to learn invariant features. We rely on the method in OCM to learn as many invariant features as possible to achieve holisticness.

## 4 OPTIMIZATION OBJECTIVE AND PROPOSED METHOD

The proposed optimization objective for holistic invariant representation learning is $\mathcal{L}_{all} = \mathcal{L}_{holistic} + \mathcal{L}_{invariant}$, where $\mathcal{L}_{holistic}$ aims to learn holistic features and $\mathcal{L}_{invariant}$ emphasizes invariance and suppresses non-invariant features. We focus only on designing $\mathcal{L}_{invariant}$ and directly use the holistic representation loss of OCM, $\mathcal{L}_{OCM}(D_t^{buf}, D_t^{new})$, as $\mathcal{L}_{holistic}$. To learn invariant features based on Definition 1, we learn features that have a positive correlation with each class regardless of the environments to form the class representation. To formalize our objective, we borrow an idea from causal inference (Jung et al., 2020), which replaces the *observational distribution* $P(Y_t^{new}|X_t^{new})$ with the *interventional distribution* $P(Y_t^{new}|do(X_t^{new}))$ in *Empirical Risk Minimization*, where $do(X)$ removes the environmental effects from the prediction of $Y$. Then the *Interventional Empirical Risk* is written as (Wang et al., 2022b):

$$\begin{aligned} \hat{R}(D_t^{new}) &= \mathbb{E}_{x \sim P(X_t^{new}), y \sim P(Y_t^{new}|do(X_t^{new}))} \mathcal{L}_{ce}(\sigma_\phi(f_\theta(x), y)) \\ &= \sum_y \sum_x \sum_z \mathcal{L}_{ce}(\sigma_\phi(f_\theta(x), y)) P(y|x, z) P(z) P(x) \end{aligned} \quad (3)$$

where $z$ is a sample from the ideal environment variable $Z$ that includes all possible environments. For a replay-based method, the interventional empirical risk is $\hat{R}(D_t^{new}) + \hat{R}(D_t^{buf})$. Optimizing this loss is equivalent to making the model ignore variant features and learn invariant features defined by Definition 1 because features that only have a positive correlation with the label in a few environments are filtered out by the operation $do(X)$ and the model only uses features that have positive correlation with the label across all environments (not influenced by specific environments) to form the representation. However, directly optimizing the interventional empirical risk is hard as $z$ is hard to observe or to annotate. In the following, we use Eq. 3 to guide the design of our method.

### 4.1 CREATING MORE ENVIRONMENTS $z$ FOR LEARNING INVARIANT FEATURES

We propose a method here to make the model learn invariant features by aligning the inputs with the same semantic meaning (class label) from different environments. The method has two parts. The first part creates augmented inputs with different environments from the original input. The augmented inputs have the same class label as the original input but diverse variant features or environments. The second part is a new optimization objective for the model to better learn invariant features across the augmented inputs and filter out variant ones based on the definition in Eq. 2. Unlike most existing augmentations that are designed for learning rich features or other purposes, our augmentations are specifically designed for learning invariant features.

**Choice of Data Augmentation.** Different from domain generalization, CL does not provide a fixed number visually distinct domains/environments or any prior knowledge about the domains (e.g., the domains/environments where the objects to be classified reside in). To achieve a similar effect of different domains or environments, we propose two data augmentations for the input image $x$ to create augmented images with different $z$'s.

**(1). *Color change*.** For an input image $x$, we first randomly reorder the RGB channels of the image to create an image $x'$ and then sample a $\lambda$ from Beta distribution to create an augmented image,

$$\hat{x} = \lambda \cdot x + (1 - \lambda) \cdot x' \quad (4)$$

We use this augmentation to create augmented images with different color environments (Fig. 1(b)). The augmented image $x'$ has the same label as $x$. We denote this augmentation as $Aug_{color}(x, s)$, where $s$ means that we repeat the two steps $s$ times to create $s$ different augmented images.

**(2). *Adding more environments*.** For input $x$ of size $d$, we first resize $x$ to $r_1 \cdot d$ where $r_1$ is the resize rate. We name this resized image as $resize(x, r_1 \cdot d)$. Then we randomly sample anther image $x_2$ from the batch $X_t^{buf} \bigcup X_t^{new}$ and replace the center part of $x_2$ with $resize(x, r_1 \cdot d)$. We denote the augmented image as $Aug_{plus}(x)$, whose original object is not changed dramatically with a reasonable choice of $r_1$. But the background change introduces features from image $x_2$ (see Fig. 1(c)).

**New Optimization Objective.** To make the model focus on invariant features, we force the representations of the augmented images to be near to each other, which penalizes the model if it uses simple color features ($Aug_{color}$) or features related to fixed backgrounds ($Aug_{plus}$) to form the class

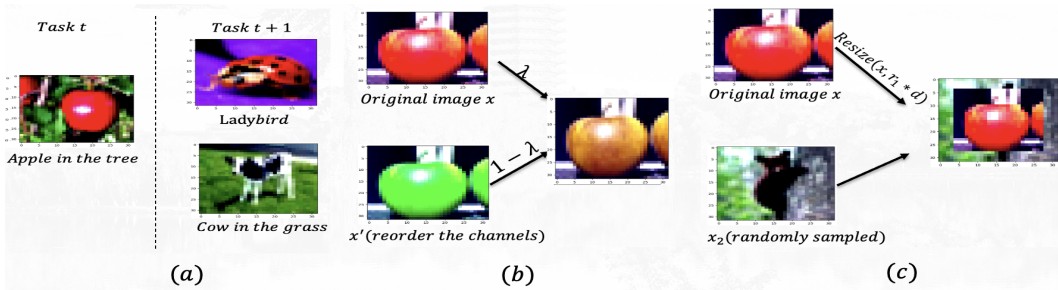

Figure 1: (a) is the illustration for the example in the Introduction and (b) is the illustration for the data augmentation $Aug_{color}$ and (c) is the illustration for the data augmentation $Aug_{plus}$.

representation (Definition 1). As we consider classification tasks, we use the two types of augmentations separately to create augmented images of $x$ as $Aug(x) = Aug_{plus}(x) \bigcup Aug_{color}(x, s)$, and calculate the cross-entropy loss of the union of the augmented images ($s + 1$ images) and the original image $x$ (1 image), The overall risk is turned into:

$$R(x, y) = \frac{\sum\limits_{x_i \in Aug(x) \bigcup\{x\}} \mathcal{L}_{ce}(x_i, y)}{s+2} + \sum_{x_i, x_j \in Aug(x) \bigcup\{x\}} \frac{dist(f_\theta(x_i), f_\theta(x_j))}{\frac{(s+2)\cdot(s+3)}{2}} \tag{5}$$

where $x_i, x_j$ are images in $Aug(x) \bigcup\{x\}$ ($x_i \neq x_j$) and $dist(\cdot, \cdot)$ computes the cosine distance between the two representations. With the regularization of the second term, $F$ learns invariant features across different environments and ignore those that only have strong positive correlations with the class under one/few environments (Eq.2). We further compare the proposed data augmentations with other related augmentations and analyze the new optimization objective in Sec. 5.4.

*Theoretical justification*: We construct an environment set $\mathbb{Z}$ and want the model to learn invariant features across different environments. However, as the number of constructed environments is limited, the model may memorize each environment and use environment-related features to reduce the loss. By optimizing the second term in Eq. 5, we force the representations of different augmented images of the original image to be very similar. Then the learner will learn little information about $\mathbb{Z}$ from the augmented images, which means $P(\mathbb{Z}|x) \approx P(\mathbb{Z})$, where $x \in Aug(x)$. Then we have

$$\hat{R}(x, y) = \sum_y \sum_x \sum_z \mathcal{L}_{ce}(\sigma_\phi(f_\theta(x)), y)P(y|x, z)P(z)P(x)$$

$$\approx \sum_y \sum_x \mathcal{L}_{ce}(\sigma_\phi(f_\theta(x)), y)P(x, y) = \frac{\sum\limits_{x_i \in Aug(x) \bigcup\{x\}} \mathcal{L}_{ce}(x_i, y)}{s+2} \tag{6}$$

So minimizing the risk in Eq. 5 is approximated by minimizing the ideal interventional empirical risk under the environments variable $\mathbb{Z}$. Note, the constructed $\mathbb{Z}$ is a subset of the ideal environment set $Z$. Increasing the diversity of samples in $\mathbb{Z}$ makes it closer to the ideal set $Z$.

## 4.2 STORING MORE DATA AND DEALING WITH LOCAL SAMPLING BIAS

Data augmentation can reduce the influence of environments only partially. For example, Samoyed dog and Golden Retriever are dogs in different environments and it's hard to use the data augmentation to generate images of Samoyed dog when we only have the images of Golden Retriever. But training $F$ with more diverse images of objects of a class can make the model learn more invariant features. So our method also focuses on better utilizing the original data to learn invariant features across different environments. First of all, OCM (Guo et al., 2022) discards the new data batch $D_t^{new}$ and uses only the buffer batch $D_t^{buf}$ (which also contains some saved new task data) to calculate $\mathcal{L}_{ce}$ loss, which makes the model use less data to learn. We add $D_t^{new}$ into our empirical risk loss,

$$R(D_t^{new} \bigcup D_t^{buf}) = \sum_{x \in X_t^{buf}} \frac{R(x, y)}{N^{buf}} + \sum_{x \in X_t^{new}} \frac{R(x, y)}{N^{new}} \tag{7}$$

where $N^{new}$ and $N^{buf}$ are the batch sizes of $D_t^{new}$ and $D_t^{buf}$ respectively. Following (Ahn et al., 2021), to avoid the interference from the new data to the replayed data, for the second term in Eq. 7, we clip the logits of classes that do not appear in $Y_t^{new}$ when calculating the cross-entropy loss.

**Resizing and storing more data:** Due to the limited buffer size, only a small number of samples are stored. However, storing more samples helps learn more invariant features because more images also mean more diversity of the environments. Without increasing the memory size, we propose to store a fixed rate (storage rate) of resized images.[4] Specifically, we use a quarter of the original memory space to store resized images. The size of a resized image is set as $r_2 \cdot d$ where $r_2$ is the resized rate and $d$ is the size of the original image. When we sample data from the buffer, we resize the sampled resized images to their original size and mix them with other sampled images to construct $X_t^{buf} = \bigcup_{i=1}^{N^{buf}} resize(x_i, d)$, where $x_i$ is an image sampled from the memory buffer. Note that Wang et al. (2022a) used data compression in offline CL to store more data by solving an optimization objective for $r_2$. But it needs the whole training set of the task and the converged model parameters after the training of the task. Those are not available in online CL. In Appendix 1, we justify the use of more samples to enable $F$ to learn better invariant feature representations and to make the risk in Eq. 7 better approximate the ideal objective in Eq. 3. Now, our $\mathcal{L}_{invariant}$ loss is

$$\mathcal{L}_{invariant}(D_t^{new}, D_t^{buf}) = \sum_{x \in \bigcup_{i=1}^{N^{buf}} resize(x_i, d_{ori})} \frac{R(x,y)}{N^{buf}} + \sum_{x \in X_t^{new}} \frac{R(x,y)}{N^{new}} \tag{8}$$

The optimization $\mathcal{L}_{all}$ loss for learning task $t$ then is

$$\mathcal{L}_{all}(D_t^{new}, D_t^{buf}) = \mathcal{L}_{OCM}(D_t^{new}, D_t^{buf}) + \mathcal{L}_{invariant}(D_t^{new}, D_t^{buf}) \tag{9}$$

where the first term learns holistic feature representations and the second term learns invariant class representations to establish the class decision boundaries. We call this method **IFO**.

**Local sampling bias (LSB).** In online CL, the model only gets a new data batch of a task each time. It is easy to learn variant features if similar environments appear in a few recent data batches. We call this *local sampling bias* (LSB). To mitigate this problem, we propose to use the buffer data that has the same label as the new data to augment the new data. Specifically, for each sample $x$ in $D_t^{new}$, we first calculate the cosine distance between $f_\theta(x)$ and the hidden representations of all stored data of the same label $y$. Then we choose the stored data sample that has the maximum representation distance with $x$ and denote it as $\dot{x}$. We don't randomly choose a stored data sample as we want $\dot{x}$ to be very different from $x$. We use $\dot{x}$ as $x_2$ in the process of creating $Aug_{plus}(x)$. Then the augmented image $Aug_{plus}(x)$ has the same object of $x$ but a new environment that is totally different from $x$. Optimizing Eq. 8 helps the model to get rid of the false high correlation between variant features that appear in recent new data batches and the class label.

Additionally, we calculate the cosine distance between $f_\theta(x)$ and the hidden representations of all buffer data that do not belong to class $y$ (of $x$). Then we choose the buffer sample that has the minimum representation distance with $x$ and denote it as $\ddot{x}$. $\ddot{x}$ has similar features/environment to $x$ but a totally different class. We collect $\ddot{x}$ for each $x$ in batch $D_t^{new}$ to construct a batch $D_t^{new,buf}$ that has the same size as $D_t^{new}$. Adding the loss $\sum_{x \in X_t^{new, buf}} \frac{R(x,y)}{N^{new}}$ into the original invariant loss (Eq 8) makes the model learn to distinguish new data and data from other classes under similar environments and also to avoid the LSB problem. We denote the new invariant loss as $\mathcal{L}_{invariant}(D_t^{new}, D_t^{buf}, D_t^{new,buf})$. Considering the LSB problem, we obtain a new model called **IFO++**, which optimizes

$$\mathcal{L}_{all}(D_t^{new}, D_t^{buf}) = \mathcal{L}_{OCM}(D_t^{new}, D_t^{buf}) + \mathcal{L}_{invariant}(D_t^{new}, D_t^{buf}, D_t^{new,buf}) \tag{10}$$

## 5 EXPERIMENT RESULTS

We evaluate the proposed IFO in three online CL scenarios: standard *disjoint task* scenario, *blurry task boundary* scenario, and *data environment shift* scenario.

### 5.1 DISJOINT ONLINE CONTINUAL LEARNING SCENARIO

**Datasets.** In this scenario, we use five datasets. For MNIST (LeCun et al., 1998), we split its 10 classes into 5 different tasks, 2 classes per task. For CIFAR10 (Krizhevsky & Hinton, 2009), we also split the 10 classes into 5 different tasks with two classes per task. For CIFAR100 (Krizhevsky

---

[4]Note that this is not image compression (Wang et al., 2022a), which should also work but more complex.

& Hinton, 2009), we split the 100 classes into 10 different tasks with 10 classes per task. For TinyImagenet (Le & Yang, 2015), we split the 200 classes into 100 different tasks with 2 classes per task for stress testing. For ImageNet (Deng et al., 2009), we split the 1000 classes int into 10 different tasks with 100 classes per task. Each task runs with only one epoch for online CL.

**Baselines.** See the 12 baselines in column 1 of Table 1. We run their official codes (Appendix 2).

**Backbone, Optimizer, Buffer and Data Augmentation.** We follow (Guo et al., 2022) and use ResNet-18 (not pre-trained) as the backbone for our method and baselines in the CIFAR10, CIFAR100, TinyImageNet, and ImageNet settings. A fully-connected network with two hidden layers (400 ReLU units) is used as the backbone for MNIST. We use the Adam optimizer and set the learning rate as 1e-3 for all methods and set $N^{new}$ and $N^{buf}$ as 10 and 64 respectively for all methods and use the reservoir sampling for our method. For the buffer size $B$, we use $\frac{B}{4}$ to store resized images and $\frac{3B}{4}$ to store the original-sized images. As the numbers of resized images and stored original-sized images are similar, if the training iteration is an even number, we store resized images and otherwise we store original-sized images. For other baselines, only the original-sized images are stored. The whole memory size is the same for all methods. Following (Guo et al., 2022), the data augmentation methods *horizontal-flip*, *random-resized-crop* and *random-gray-scale* are applied to all methods to improve the performance (no drops). Note that these augmentations have also been used in other papers, e.g., Fini et al. (2020). We set $s$ in Eq. 5 as 5 and $r_1$ in $Aug_{plus}$ as 0.75 and $r_2$ as 0.5. More details are in Appendix 2.

Table 1: Accuracy on the MNIST (5 tasks), CIFAR10 (5 tasks), CIFAR100 (10 tasks) and TinyImageNet (100 tasks) datasets with different memory buffer sizes $B$. All values are the averages of 15 runs. See the results on ImageNet in Figure 2(a).

| Method | MNIST | | | CIFAR10 | | | CIFAR100 | | | TinyImageNet | | |
|---|---|---|---|---|---|---|---|---|---|---|---|---|
| $B$ | B=0.1k | B=0.5k | B=1k | B=0.2k | B=0.5k | B=1k | B=1k | B=2k | B=5k | B=2k | B=4k | B=10k |
| AGEM Chaudhry et al. (2018) | 56.9±5.2 | 57.7±8.8 | 61.6±3.2 | 22.7±1.8 | 22.7±1.9 | 22.6±0.7 | 5.8±0.2 | 5.8±0.3 | 6.5±0.2 | 0.9±0.1 | 2.1±0.1 | 3.9±0.2 |
| GSS Aljundi et al. (2019b) | 70.4±1.5 | 80.7±5.8 | 87.5±5.9 | 26.9±1.2 | 30.7±1.3 | 40.1±1.4 | 11.1±0.2 | 13.3±0.5 | 17.4±0.1 | 3.3±0.5 | 10.0±0.2 | 10.5±0.2 |
| ER Chaudhry et al. (2020) | 78.7±0.4 | 88.0±0.2 | 90.3±0.1 | 29.7±1.0 | 35.2±0.3 | 44.3±0.4 | 11.7±0.3 | 15.0±0.9 | 14.4±0.9 | 5.6±0.5 | 10.1±0.7 | 11.7±0.2 |
| MIR Aljundi et al. (2019a) | 79.0±0.5 | 88.3±0.1 | 91.3±1.9 | 37.3±0.3 | 40.0±0.6 | 41.0±0.6 | 15.7±0.2 | 19.1±0.1 | 24.1±0.2 | 6.1±0.5 | 11.7±0.2 | 13.5±0.2 |
| ASER Shim et al. (2021) | 61.6±2.1 | 71.0±0.6 | 82.1±5.9 | 27.8±1.0 | 36.2±1.2 | 44.7±1.2 | 16.4±0.3 | 12.2±1.9 | 27.1±0.3 | 5.3±0.3 | 8.2±0.2 | 10.3±0.4 |
| GDumb Prabhu et al. (2020) | 81.2±0.5 | 91.0±0.2 | 94.5±0.1 | 35.9±1.1 | 50.7±0.7 | 63.5±0.5 | 14.1±0.3 | 20.1±0.2 | 36.0±0.5 | 12.6±0.1 | 12.7±0.3 | 15.7±0.2 |
| SCR Mai et al. (2021) | 86.2±0.5 | 92.8±0.3 | 94.6±0.1 | 47.2±1.7 | 58.2±0.5 | 64.1±1.2 | 26.5±0.2 | 31.6±0.5 | 36.5±0.2 | 10.6±1.1 | 17.2±0.1 | 20.4±1.1 |
| DER++ Buzzega et al. (2020) | 74.4±1.1 | 91.5±0.2 | 92.1±0.2 | 44.2±1.1 | 47.9±1.5 | 54.7±2.2 | 15.3±0.2 | 19.7±1.5 | 27.0±0.7 | 4.5±0.3 | 10.1±0.3 | 17.6±0.5 |
| IL2A Zhu et al. (2021) | 90.2±0.1 | 92.7±0.1 | 93.9±0.1 | 54.7±0.5 | 56.0±0.4 | 58.2±1.2 | 18.2±1.2 | 19.7±0.5 | 22.4±0.2 | 5.5±0.7 | 8.1±1.2 | 11.6±0.4 |
| Co$^2$L Cha et al. (2021) | 83.1±0.1 | 91.5±0.1 | 94.7±0.1 | 42.1±1.2 | 51.0±0.7 | 58.8±0.4 | 17.1±0.4 | 24.2±0.2 | 32.2±0.5 | 10.1±0.2 | 15.8±0.4 | 22.5±1.2 |
| SSIL Ahn et al. (2021) | 88.2±0.1 | 93.0±0.2 | 95.1±0.1 | 49.5±0.2 | 59.2±0.4 | 64.0±0.5 | 26.0±0.1 | 33.1±0.5 | 39.5±0.4 | 9.6±0.7 | 15.2±1.5 | 21.1±0.1 |
| OCM Guo et al. (2022) | 90.7±0.1 | 95.7±0.3 | 96.7±0.1 | 59.4±0.2 | 70.0±1.3 | 77.2±0.5 | 28.1±0.3 | 35.0±0.4 | 42.4±0.5 | 15.7±0.2 | 21.2±0.4 | 27.0±0.3 |
| IFO | **92.5**±0.4 | **96.1**±0.2 | **97.0**±0.2 | **65.0**±0.3 | **73.5**±0.2 | **78.0**±0.3 | **38.5**±0.5 | **45.5**±0.3 | **49.4**±0.2 | **20.5**±0.5 | **27.2**±0.5 | **34.6**±0.4 |
| IFO++ | **93.0**±0.2 | **96.4**±0.2 | **97.0**±0.2 | **69.5**±0.2 | **76.4**±0.5 | **78.5**±0.4 | **40.5**±0.4 | **47.9**±0.7 | **51.3**±0.4 | **21.7**±0.3 | **28.9**±0.1 | **35.8**±0.2 |

**Accuracy results.** We report the average accuracy of all tasks after learning of the final task in Table 1. We observe that our IFO outperforms all baselines by a large margin. Our main baseline is OCM as we added the condition of *invariance* to its *holistic representation* learning. IFO boosts the performance of OCM significantly. IFO's performance is especially strong when the buffer size is small (e.g., 10 samples per class). The reason is that the baselines tend to overfit the buffer data and learn variant features when the buffer size is small, which IFO is able to avoid by learning invariant features. Further, the improvement of IFO does not decrease when the dataset gets more complex (e.g., TinyImageNet). IFO++ further boosts the performance by considering *local sampling bias.*

For the hardest ImageNet dataset, due to the poor overall performance, we compare IFO with top-3 baselines (OCM, SSIL, and SCR). Figure 2(a) shows the average accuracy of all tasks seen so far after learning each task. IFO outperforms the three baselines in the whole learning process. Also, the average accuracy first arises and then drops. This is because the random-initialized model doesn't have enough features to solve the first task until the second task arrives. The later drop is due to CF.

**Forgetting rate.** We report the average forgetting rate (Chaudhry et al., 2020) in Table A.3 in Appendix 3. We see that IFO gives less forgetting than OCM, especially on CIFAR10 and TinyImageNet. Also, IFO forgets the least except for GDumb and SCR on TinyImageNet. But the accuracy of the two baselines are much lower than IFO. For ImageNet, IFO also fares well (see Appendix 3).

**Learning invariant representations.** We use two types of experiments for this evaluation: (1) model robustness on unseen environments and (2) eigenvalue distribution of the learned representations. Due to space limits, we give the details in Appendix 4. All experiments verify the effectiveness of our method in learning invariant features.

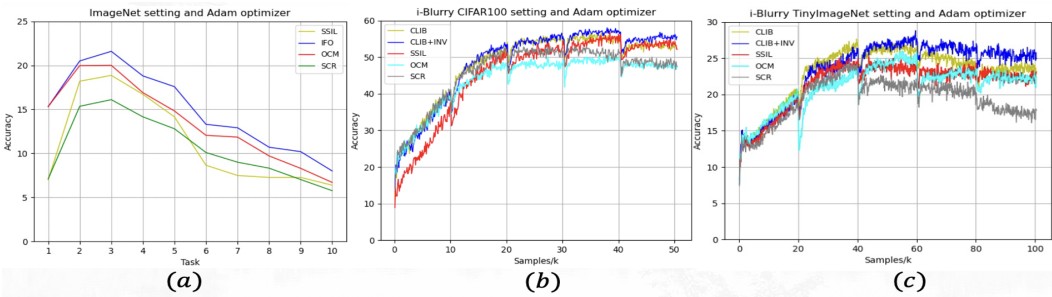

Figure 2: (a) results in the disjoint setting using ImageNet. The buffer size is 10k. For (b) and (c), 'samples' of the x-axis means the number of new data samples that the model has seen and the unit is 1k. (b) results of the online i-Blurry setting using CIFAR100. The buffer size is 2k. (c) results of the online i-Blurry setting using TinyImageNet. The buffer size is 4k.

## 5.2 BLURRY ONLINE CONTINUAL LEARNING SCENARIO

In this scenario (Koh et al., 2021), the classes of the training data for a dataset is split into two parts, $N\%$ of the classes as the disjoint part and the rest of $100 - N\%$ of the classes as the blurry part. Each task consists of some disjoint classes (with all their data) and some blurry class samples. The disjoint classes of a task do not appear in any other task. The blurry classes data are sampled from the blurry part. Specifically, for a task, $100 - M\%$ of the samples are from some randomly selected dominant blurry classes (these classes will not be dominant classes in other tasks) and $M\%$ of the samples are from the other minor classes. This strategy of forming tasks is called *i-Blurry*.

Following (Koh et al., 2021), we use the i-Blurry setup with $N = 50$ and $M = 10$ (i-Blurry-50-10) in our experiments on the CIFAR100 (5 tasks) and TinyImageNet (5 tasks) datasets. We use ResNet-34 and Adam optimizer with an initial learning rate of 0.0003 for all systems. We compare their original system CLIB with our CLIB+INV (replacing CLIB's cross-entropy loss with our $\mathcal{L}_{invariant}$ loss in Eq. 8) and three best-performing baselines (OCM, SSIL and SCR). Here we do not use $\mathcal{L}_{OCM}$ as our main contribution is the $\mathcal{L}_{invariant}$ loss. More details is in Appendix 5.

**Accuracy results** Following (Koh et al., 2021), we measure any time inference and plot the accuracy-to-samples curve in Figure 2(b)&(c). The setting is as follows. After the model sees every 100 new samples, we test the model using the original test set of all classes in the CIFAR100 or TinyImageNet dataset and record the accuracy. The figures show that CLIB+INV clearly outperforms CLIB, especially after the model has seen some tasks, which indicates that effectively learning invariant class representations improves the overall generalization ability in incremental learning. We also observe that the three best baselines' performances are weaker than that of CLIB because the three baselines are not designed for this setting.

## 5.3 DATA ENVIRONMENT SHIFT IN ONLINE CONTINUAL LEARNING

In this scenario, the model needs to learn the same classes from different environments sequentially, and is then tested in an unseen environment. We use the PACS dataset (Li et al., 2017) to simulate this scenario. This dataset has four different environments: art painting, cartoon, photo, and sketch. Each environment has data points of the same seven classes. We choose each set of three environments as the training environments and the remaining environment as the test environment. Thus four experiments are conducted with different training and test environments. We report the average test performance over the four experiments as the empirical estimation of the ability of our method in learning invariant features. The core of this setting is to learn invariant features under the shift of environments. So we focus on the representation-learning loss and do not consider/use the replay strategy to overcome forgetting. The hyperparameters are given in Appendix 6.

Table 5 in Appendix 6 shows that the performance of our $\mathcal{L}_{invariant}$ loss (Eq. 8) is beyond that of the traditional cross-entropy loss $\mathcal{L}_{ce}$ over the four environments. This means our $\mathcal{L}_{invariant}$ loss learns invariant features better. Another observation is that optimizing the classification loss ($\mathcal{L}_{invariant}$ or $\mathcal{L}_{ce}$) with the $\mathcal{L}_{holistic}$ loss improves the performance further because learning more features of one class enables the model to learn more knowledge to deal with test samples from unseen environments.

Table 2: Ablation accuracy - average of 5 runs. $B$ is the memory buffer size.

| Dataset | no $Aug_{color}$ | no $Aug_{plus}$ | no align | OCM+Mixup | OCM+CutMix | OCM+MemoryAug | OCM+ClassAug | OCM+$X^{new}$ | no new data | no resized image |
|---|---|---|---|---|---|---|---|---|---|---|
| CIFAR100 ($B$=2$k$) | 41.8±0.4 | 44.0±0.6 | 44.6±0.2 | 33.2±0.5 | 36.3±0.3 | 35.7±0.1 | 39.5±0.4 | 37.4±0.5 | 42.0±0.2 | 44.0±0.6 |
| TinyImageNet ($B$=2$k$) | 18.0±0.3 | 19.5±0.1 | 20.0±0.3 | 14.4±0.2 | 16.0±0.5 | 17.5±0.3 | 17.7±0.5 | 23.0±0.5 | 18.5±0.6 | 19.9±0.4 |

## 5.4 ABLATION STUDY AND ANALYSIS

*Ablation study of the components in IFO.* Table 6 shows the results without using $Aug_{color}$ (no $Aug_{color}$), $Aug_{plus}$ (no $Aug_{plus}$) or the second term of Eq. 5 (no align). The performances of these experiments drop, which show the contribution of the proposed augmentations and loss. As IFO is basically OCM plus invariant feature learning, we also tried some other data augmentations and losses designed for feature learning, Mixup (Zhang et al., 2017) (OCM+Mixup)), Cutmix (Yun et al., 2019) (OCM+CutMix), MemoryAug (Fini et al., 2020) (OCM+MemoryAug) and ClassAug (Zhu et al., 2021) (OCM+ClassAug). CutMix, MemoryAug and ClassAug improve the performance of OCM but Mixup does not. However, the results of these augmentations and their losses are poorer than that of IFO. For "OCM+$X^{new}$", we add $\mathcal{L}_{ce}$ loss of the new data batch into OCM. The performance improves slightly as the model learns more invariant features from more data. For "no new data", only $D_t^{buf}$ in Eq. 8 is used. For "no resized image", we do not store resized images. The performances of both experiments drop, which means that considering more samples to learn invariant features is useful. Ablation experiments on IFO++ are discussed in Appendix 7.

*Influence of hyperparameters in IFO.* For $s$ in Eq. 5, from Figure 4(a) in Appendix 8, we observe that the performance has a positive correlation with the number $s$ as the model gradually focuses on invariant features rather than strongly depending on simple colors. We set $s$ to 5 as it achieves the best performance with less compute. For the storage rate for resized images, we need to find a balance between storing more samples and avoiding huge information loss caused by the resizing operation. From Figure 4(b), we found that using a quarter of the memory space to store resized images achieves a good balance. For rate $r_1$ in the augmentation $Aug_{plus}$, we need to avoid introducing trivial features ($r_1$ is too high) and causing a huge information loss in the original image ($r_1$ is too low). For rate $r_2$ in Sec. 4.2, we have the same balancing issue with the storage rate (more saved samples vs information loss). Based on Figure 4(c), we set $r_1$ as 0.75 and $r_2$ as 0.5.

*Influence of learning invariant features for continual learning (CL).* To further investigate how learning invariant features helps the performance of CL methods, we measure three abilities of OCM and IFO: **(1)** the ability to establish decision boundaries between the classes within the new task by recording the accuracy performance of the new task. From Figure 6(a) in Appendix 9, we see that IFO's accuracy of the new task outperforms that of OCM as learning invariant features makes the model improve its generalization power. **(2)** the ability to maintain the learned decision boundaries within a task by calculating the average task incremental accuracy of the previous tasks. Our method IFO again outperforms OCM (Figure 6(b)) as IFO mitigates the overfitting problem of the limited buffer data. **(3)** the ability to establish class boundaries across tasks. We measure this by considering only the logit of the true label of each test instance in a task and the logits of the classes from the other tasks when the model predicts the label of the test instance. IFO's performance is still better than that of OCM (Figure 6(c)) as IFO reduces the variant features, which enables the model to project those class representations to different locations in the space, making the decision boundaries easier to establish. Also from Figure 6, we can clearly see that improving the last ability is the biggest challenge for online CL. More details are given in Appendix 9.

## 6 CONCLUSION

Despite the fact that numerous empirical techniques have been proposed to solve the class-incremental learning (CIL) problem, limited work has been done to study the necessary conditions for good CIL performance. Recent work in (Guo et al., 2022) proposes that it is neccesary to learn *holistic feature representations*, which improved the online CIL performance significantly. This paper further argued that it is also necessary to learn *invariant features* for each class, and proposed several methods to help learn invariant features in addition to learning holistic representations. Experimental results showed that the new condition gave another boost to the online CIL performance.

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

# A   APPENDIX

## A.1   *Theoretical justification* FOR SECTION 4.2

In the ideal case (data is sufficient and highly diverse), $X$ can be observed in any environment $z$. That leads to $P(Z|X) \approx P(Z)$ (the independence of $Z$ and $X$) and we have

$$
\begin{aligned}
\hat{R}(D_t^{new}) &= \sum_{y \in Y_t^{new}} \sum_{x \in X_t^{new}} \sum_z \mathcal{L}_{ce}(\sigma_\phi(f_\theta(x), y)) P(y|x, z) P(z) P(x) \\
&\approx \sum_{y \in Y_t^{new}} \sum_{x \in X_t^{new}} \mathcal{L}_{ce}(\sigma_\phi(f_\theta(x), y)) P(y, x)
\end{aligned}
\tag{11}
$$

where $\sum_y \sum_x \mathcal{L}_{ce}(\sigma_\phi(f_\theta(x), y)) P(y, x)$ is the traditional cross-entropy loss. We also have $\hat{R}(X_t^{buf}, Y_t^{buf}) = \sum_{y \in Y_t^{buf}} \sum_{x \in X_t^{buf}} \mathcal{L}_{ce}(\sigma_\phi(f_\theta(x), y)) P(y, x)$. So adding $X^{new}$ and resizing and storing more buffer data to construct $X^{buf}$ makes the model have more samples from different environments to train. And that makes the risk in Eq. 7 better approximated to the ideal objective in Eq. 3.

## A.2   THE DETAILS OF DISJOINT ONLINE CONTINUAL LEARNING SCENARIO

Due to the limitation of computational resources, we download the downsampled version of ImageNet ($3 \times 32 \times 32$) from the official website and conduct experiments on this dataset. We don't use the $Aug_{color}$ in the MNIST setting. We set the alpha and the beta of the beta distribution in $Aug_{color}$ as 1.

For AGEM, following the original paper, we use the SGD optimizer and set the learning rate as 0.1. We use the random method to update the buffer and to sample data.

For GSS, based on the original paper, we use the same optimizer and learning rate as above. The number of buffer batches randomly sampled from the memory to estimate the maximal gradients cosine similarity score is set to 10 and the randomly sampled buffer batch ($X^{buffer}$) size for calculating the score is 64.

For ASER, we use the mean value of Adversarial SV and Cooperative SV, and set the maximum number of samples per class for random sampling as 1.5. We allow 3 nearest neighbors for KNN-SV computation. We use the same SV-based methods for both Memory-Update and Memory-Retrieval as given in the original paper.

For MIR/ER, we use the Adam optimizer and we set the learning rate as 0.001 and fix the weight decay as 0.0001. We set the sub-sample size as 128.

For DER++, we use the Adam optimizer, set the learning rate as 0.001, fix the weight decay as 0.0001 and the value of alpha ($\alpha$) as 0.1, and fix the beta ($\beta$) as 0.5.

For GDumb, we use the Adam optimizer, set the learning rate as 0.001 and fix the weight decay as 0.0001. We use the CutMix as the regularization to overcome over-fitting. we follow the official code and set the number of epochs for training the whole buffer data as 256 for MINIST, CIFAR10, and CIFAR100 datasets, and 32 for the TinyImagenet dataset. We set the gradient clip as 10.

For SCR, we use the Adam optimizer, set the learning rate as 0.001 and fix the weight decay as 0.0001. We set the temperature for contrastive loss as 0.07. We employ a linear layer with the size $[dim_h, 128]$ as the contrastive head. We follow the official code and use the horizontal-flip, random-resized crop, random-gray-scale,color-jitter as its data augmentations.

The official code of these systems can be found from the following locations.

The code of ER and MIR: `https://github.com/optimass/Maximally_Interfered_Retrieval`.
The code of ASER and SCR: `https://github.com/RaptorMai/online-continual-learning`.
The code of GDumb: `https://github.com/drimpossible/GDumb`.
The code of DER++: `https://github.com/aimagelab/mammoth`.
The code for AGEM: `https://github.com/facebookresearch/agem`.

The code for GSS: `https://github.com/rahafaljundi/Gradient-based-Sample-Selection`.
The code for Co²L: `https://github.com/chaht01/Co2L`.
The code for IL2A: `https://github.com/Impression2805/IL2A`.
The code for SSIL: `https://github.com/hongjoon0805/SS-IL-Official`.
The code for OCM: `https://github.com/gydpku/OCM`.

## A.3 THE FORGETTING RATE TABLE

Table 3: Average forgetting rate. All numbers are the averages of 15 runs. See the forgetting rates for the ImageNet dataset in the text below.

| Method | MNIST | | | CIFAR10 | | | CIFAR100 | | | TinyImageNet | | |
|---|---|---|---|---|---|---|---|---|---|---|---|---|
| B | B=0.1k | B=0.5k | B=1k | B=0.2k | B=0.5k | B=1k | B=1k | B=2k | B=5k | B=2k | B=4k | B=10k |
| AGEM Chaudhry et al. (2018) | $32.5_{\pm5.9}$ | $30.1_{\pm4.2}$ | $32.0_{\pm2.9}$ | $36.1_{\pm3.8}$ | $43.2_{\pm4.3}$ | $48.1_{\pm3.4}$ | $43.3_{\pm0.2}$ | $45.7_{\pm0.3}$ | $43.9_{\pm0.2}$ | $73.9_{\pm0.2}$ | $78.9_{\pm0.2}$ | $74.1_{\pm0.3}$ |
| GSS Aljundi et al. (2019b) | $26.1_{\pm2.2}$ | $17.8_{\pm5.22}$ | $10.5_{\pm6.7}$ | $75.5_{\pm1.5}$ | $65.9_{\pm1.6}$ | $54.9_{\pm2.0}$ | $30.8_{\pm0.2}$ | $30.7_{\pm0.5}$ | $26.4_{\pm0.3}$ | $72.8_{\pm1.2}$ | $72.6_{\pm0.4}$ | $71.5_{\pm0.2}$ |
| ER Chaudhry et al. (2020) | $22.7_{\pm0.5}$ | $9.7_{\pm0.4}$ | $6.7_{\pm0.5}$ | $42.0_{\pm0.5}$ | $26.7_{\pm0.7}$ | $20.7_{\pm0.7}$ | $34.2_{\pm0.2}$ | $31.7_{\pm0.9}$ | $35.3_{\pm0.9}$ | $68.2_{\pm2.8}$ | $66.2_{\pm0.8}$ | $67.2_{\pm0.2}$ |
| MIR Aljundi et al. (2019a) | $22.3_{\pm0.5}$ | $9.0_{\pm0.5}$ | $5.7_{\pm0.9}$ | $40.0_{\pm1.6}$ | $25.9_{\pm0.7}$ | $24.5_{\pm0.5}$ | $24.5_{\pm0.3}$ | $21.4_{\pm0.3}$ | $21.0_{\pm0.1}$ | $61.1_{\pm3.2}$ | $60.9_{\pm0.3}$ | $59.5_{\pm0.3}$ |
| ASER Shim et al. (2021) | $33.8_{\pm1.1}$ | $24.8_{\pm0.5}$ | $13.8_{\pm0.4}$ | $71.1_{\pm1.8}$ | $59.1_{\pm1.5}$ | $50.4_{\pm1.5}$ | $25.0_{\pm0.2}$ | $12.2_{\pm1.9}$ | $13.2_{\pm0.1}$ | $65.7_{\pm0.7}$ | $64.2_{\pm0.2}$ | $62.2_{\pm0.1}$ |
| GDumb Prabhu et al. (2020) | $10.3_{\pm0.1}$ | $6.2_{\pm0.1}$ | $4.8_{\pm0.2}$ | $26.5_{\pm0.5}$ | $24.5_{\pm0.2}$ | $18.9_{\pm0.4}$ | $16.7_{\pm0.5}$ | $17.6_{\pm0.2}$ | $16.8_{\pm0.4}$ | $15.9_{\pm0.5}$ | $14.6_{\pm0.3}$ | $11.7_{\pm0.2}$ |
| SCR Mai et al. (2021) | $10.7_{\pm0.1}$ | $4.7_{\pm0.1}$ | $4.0_{\pm0.2}$ | $41.3_{\pm0.1}$ | $31.5_{\pm0.2}$ | $24.7_{\pm0.4}$ | $17.5_{\pm0.2}$ | $11.6_{\pm0.5}$ | $5.6_{\pm0.4}$ | $19.4_{\pm0.3}$ | $15.4_{\pm0.3}$ | $14.9_{\pm0.7}$ |
| DER++ Buzzega et al. (2020) | $25.0_{\pm0.3}$ | $7.3_{\pm0.3}$ | $6.6_{\pm1.2}$ | $30.1_{\pm0.8}$ | $31.8_{\pm2.5}$ | $18.7_{\pm3.4}$ | $43.4_{\pm0.2}$ | $44.0_{\pm1.9}$ | $25.8_{\pm3.5}$ | $67.2_{\pm1.7}$ | $63.6_{\pm0.3}$ | $55.2_{\pm0.7}$ |
| IL2A Zhu et al. (2021) | $8.7_{\pm0.1}$ | $7.2_{\pm0.1}$ | $4.1_{\pm0.1}$ | $36.0_{\pm0.2}$ | $32.1_{\pm0.4}$ | $29.1_{\pm0.4}$ | $24.6_{\pm0.6}$ | $12.5_{\pm0.7}$ | $20.0_{\pm0.5}$ | $65.5_{\pm0.7}$ | $60.1_{\pm0.5}$ | $57.6_{\pm1.1}$ |
| Co²L Cha et al. (2021) | $14.7_{\pm0.2}$ | $7.1_{\pm0.1}$ | $3.1_{\pm0.1}$ | $32.0_{\pm0.1}$ | $21.0_{\pm0.3}$ | $16.9_{\pm0.2}$ | $16.9_{\pm0.4}$ | $16.6_{\pm0.6}$ | $9.9_{\pm0.7}$ | $60.5_{\pm0.5}$ | $52.5_{\pm0.9}$ | $42.5_{\pm0.8}$ |
| SSIL Ahn et al. (2021) | $11.3_{\pm0.1}$ | $2.7_{\pm0.1}$ | $2.8_{\pm0.1}$ | $36.0_{\pm0.7}$ | $29.6_{\pm0.4}$ | $13.5_{\pm0.4}$ | $40.1_{\pm0.5}$ | $33.9_{\pm1.2}$ | $21.7_{\pm0.8}$ | $44.4_{\pm0.7}$ | $36.6_{\pm0.7}$ | $29.0_{\pm0.7}$ |
| OCM Guo et al. (2022) | $4.7_{\pm0.1}$ | $1.8_{\pm0.1}$ | $1.3_{\pm0.1}$ | $23.0_{\pm0.2}$ | $14.0_{\pm0.7}$ | $12.0_{\pm1.1}$ | $12.2_{\pm0.3}$ | $8.5_{\pm0.3}$ | $4.5_{\pm0.3}$ | $23.5_{\pm1.9}$ | $21.0_{\pm0.3}$ | $18.6_{\pm0.5}$ |
| IFO | $4.2_{\pm0.1}$ | $1.0_{\pm0.2}$ | $1.1_{\pm0.1}$ | $16.3_{\pm0.5}$ | $8.1_{\pm0.1}$ | $2.0_{\pm0.7}$ | $11.9_{\pm0.5}$ | $8.3_{\pm0.4}$ | $4.3_{\pm0.2}$ | $22.5_{\pm0.3}$ | $18.6_{\pm0.5}$ | $13.5_{\pm0.8}$ |
| IFO++ | $3.0_{\pm0.1}$ | $0.8_{\pm0.2}$ | $0.9_{\pm0.1}$ | $12.7_{\pm0.5}$ | $6.1_{\pm0.2}$ | $2.0_{\pm0.3}$ | $9.8_{\pm0.5}$ | $7.3_{\pm0.4}$ | $3.3_{\pm0.5}$ | $21.5_{\pm0.3}$ | $15.9_{\pm0.2}$ | $12.9_{\pm0.2}$ |

From this forgetting table (Table A.3), we observe an obvious drop in forgetting rate from OCM to IFO and IFO++ (our methods). That means our methods forget less. In the ImageNet setting, the forgetting rates for the four top methods are 11.47 (SCR), 12.1 (SSIL), 11.7 (OCM) and 10.9 (IFO).

## A.4 LEARNING INVARIANT REPRESENTATIONS.

Although better results of our method in the disjoint (Sec.5.1), blurry (Sec.5.2), and data environment shift (Sec. 5.3) experiments have already indicated that our proposal is able to learn invariant features better. Here we use two additional and more specific types of experiments to further evaluate the effectiveness of learning invariant features: (1) model robustness on unseen environments and (2) eigenvalue distribution of the learned representations.

**(1) Model robustness on unseen environments**. To verify that our method has learned invariant features to form the class representation, we conduct the following experiment (this is not a continual learning setting). After training on the full CIFAR100 data, we test the trained model on the CIFAR100-C dataset (Hendrycks & Dietterich, 2019), which is a model robustness benchmark consisting of 19 corruption types with five levels of severities applied to the original test set of CIFAR100. The corruptions come from four main categories: noise, blur, weather, and digital. Each corruption has five-level severities and "5" indicates the most corrupted one. Those corruptions are not used in the training, so the test can be viewed as a test of the trained model in unseen environments. A model that has learned invariant features should achieve a higher performance. From Table 4, we observe that IFO indeed outperforms the OCM method on 19 unseen environments. The gap between IFO and OCM gets larger with the level of severity increases. We conduct a similar experiment on the Tiny-ImageNet-C dataset (another robustness dataset in (Hendrycks & Dietterich, 2019)), and the conclusion is consistent (Table 4). Those experiments empirically verify that our method learns more invariant features than OCM.

**(2) eigenvalue distribution of the learned representations.** Zhu et al. (2021) observed that representations $f_\theta(x)$ with larger eigenvalues transfer better and suffer less forgetting. Guo et al. (2022) showed the number of eigenvectors with significant variances is a good indicator of holistic degree of the learned representations. We follow their experiment setting and plot the eigenvalues of the eigenvectors of the representations learned with ER, OCM and IFO from the first 50 classes of CIFAR-100 as a single supervised learning task (see Appendix 4). From Figure 3, we see that IFO and OCM have much more significant directions (eigenvectors) than the basic ER method (Chaudhry et al., 2020), which shows that both IFO and OCM have learned more diverse/holistic representations.

Table 4: Test accuracy in the robustness benchmark. All numbers are the averages of 15 runs

| Dataset | CIFAR100-C | | | | | Tiny-ImageNet-C | | | | |
|---|---|---|---|---|---|---|---|---|---|---|
| Methods | Severity 1 | Severity 2 | Severity 3 | Severity 4 | Severity 5 | Severity 1 | Severity 2 | Severity 3 | Severity 4 | Severity 5 |
| OCM | 38.5±0.2 | 34.9±0.2 | 32.0±0.4 | 28.9±0.7 | 24.1±0.3 | 13.7±0.2 | 11.6±0.5 | 9.0±0.3 | 6.7±0.3 | 5.0±0.3 |
| IFO | 46.5±0.2 | 42.5±0.3 | 39.4±0.5 | 35.9±0.4 | 30.1±0.7 | 14.3±0.3 | 12.1±0.3 | 9.6±0.2 | 7.1±0.2 | 5.6±0.2 |

Our IFO method has a similar number of significant directions to that of OCM, but the eigenvalues of the eigenvectors of IFO and OCM are very different (see Appendix 4), which gives us indirect evidence that IFO is able to emphasize more or less of some directions compared to OCM. Coupled with IFO's better result, we can conclude that IFO finds better features.

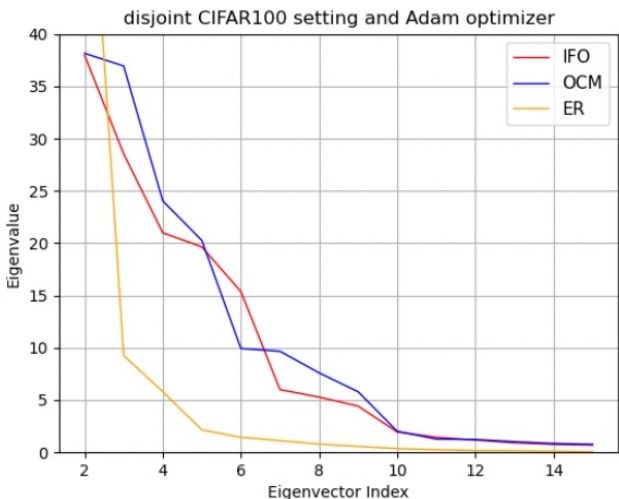

Figure 3: We plot the Eigenvalue distribution of the learned representations by ER, OCM, and IFO using the CIFAR100 dataset.

From Figure 3, we see that the main difference of the eigenvalue distributions of OCM and IFO is the different eigenvalues they learned in the first 10 eigenvectors. That means that the representation quality not only depends on the diversity of eigenvectors (holistic), but also on the different emphases (different eigenvalues) on the features for class representations to achieve invariance.

## A.5 MORE DETAILS OF I-BLURRY ONLINE CONTINUAL LEARNING SCENARIO

Following (Koh et al., 2021), for all method, we use the batch size of 16 and 3 updates per streamed sample for CIFAR100 and the batch size of 32 and 3 updates per streamed sample for TinyImageNet, and employ ResNet-34 for CIFAR100 and TinyImageNet. In the CIFAR100 setting, each task has 10 unique disjoint classes and 10 dominant blurry classes exclusively. The selection process of classes for each task is random. In the TinyImageNet setting, each task has 20 unique disjoint classes and 20 dominant blurry classes exclusively. AutoAugment (Cubuk et al., 2019) and CutMix (Yun et al., 2019) are also used as data augmentations. For CLIB and CLIB+INV, we use the same adaptive learning rate schedule (Koh et al., 2021) with $\gamma = 0.95$ and $m = 10$ for the two datasets. We use their official code https://github.com/naver-ai/i-Blurry to run the experiments.

## A.6 DETAILS OF DATA ENVIRONMENT SHIFT ONLINE CONTINUAL LEARNING SCENARIO

We use ResNet-18 (not pre-trained) as the backbone for our method and baselines and use the Adam optimizer and set the learning rate as 1e-3 for all methods. The batch size for the new data batch is 10 and there is no need to store or sample buffer data [bing: why?]. Data of each environment is run by one epoch. From Table 5, we observe that the model optimized with our invariance loss achieves higher test performance than that of the traditional cross-entropy loss in the unseen environment.

## A.7 ABLATION STUDY ON IFO++

Table 5: Test accuracy in the unseen environment. All numbers are the averages of 15 runs. 'Art-painting' means that the model first learns the other three environments sequentially (order: *Cartoon* → *Photo* → *Sketch*), then it is tested on the data points of the environment 'Art-painting'. 'Cartoon' means that the model first learns the other three environments sequentially (order: *Art-Painting* → *Photo* → *Sketch*), then it is tested on the data points of the environment 'Cartoon'. 'Photo' means that the model first learns the other three environments sequentially (order: *Art-Painting* → *Cartoon* → *Sketch*), then it is tested on the data points of the environment 'Photo'. 'Sketch' means that the model first learns the other three environments sequentially (order: *Art-Painting* → *Cartoon* → *Photo*), then it is tested on the data points of the environment 'Sketch'.

| Method | Art-painting | Cartoon | Photo | Sketch |
|---|---|---|---|---|
| $\mathcal{L}_{ce}$ | $12.7_{\pm 0.3}$ | $15.8_{\pm 0.5}$ | $12.5_{\pm 0.2}$ | $12.1_{\pm 0.3}$ |
| $\mathcal{L}_{invariant}$ | $13.5_{\pm 0.2}$ | $17.4_{\pm 0.3}$ | $13.3_{\pm 0.4}$ | $13.2_{\pm 0.6}$ |
| $\mathcal{L}_{holistic} + \mathcal{L}_{ce}$ | $14.7_{\pm 0.5}$ | $17.8_{\pm 0.2}$ | $13.2_{\pm 0.6}$ | $13.1_{\pm 0.4}$ |
| $\mathcal{L}_{holistic} + \mathcal{L}_{invariant}$ | $\mathbf{15.0_{\pm 0.4}}$ | $\mathbf{18.9_{\pm 0.2}}$ | $\mathbf{14.6_{\pm 0.1}}$ | $\mathbf{17.2_{\pm 0.4}}$ |

Table 6: Ablation accuracy - average of 5 runs. $B$ is the memory buffer size.

| Dataset | IFO+random | IFO+min | IFO+random $D_t^{new,buf}$ | IFO+max | IFO+$D_t^{new,buf}$ |
|---|---|---|---|---|---|
| CIFAR100 ($B$=2k) | $45.6_{\pm 0.2}$ | $46.0_{\pm 0.4}$ | $44.0_{\pm 0.3}$ | $46.8_{\pm 0.3}$ | $47.0_{\pm 0.3}$ |
| TinyImageNet ($B$=2k) | $20.0_{\pm 0.2}$ | $19.7_{\pm 0.4}$ | $20.6_{\pm 0.5}$ | $20.9_{\pm 0.4}$ | $21.0_{\pm 0.3}$ |

IFO++ goes further than IFO to mitigate the *local sampling bias* (LSB) problem in online CL. We report its ablation results in Table 6. In experiments 'IFO+random' and 'IFO+min', we randomly sample a buffer data as $\dot{x}$ and choose the data sample with the minimum distance from $x$ as $\dot{x}$, respectively. Experimental results show that their performances are similar or even poorer than that of IFO. This is because their $\dot{x}$ may not provide a new environment for $x$ to create its augmented image $Aug_{plus}(x)$. The performance of randomly choosing $\ddot{x}$ to construct $D_t^{new,buf}$ (IFO+random $D_t^{new,buf}$) is poorer than that of IFO++ because IFO++ requires the model to distinguish $D_t^{new}$ and $D_t^{new,buf}$ sampled from a similar environment to $D_t^{new}$, which penalizes the local sampling bias. In experiments 'IFO+max' and 'IFO+$D_t^{new,buf}$', we only consider $\dot{x}$ to augment $x$ and $\ddot{x}$ to construct $D_t^{new,buf}$ and calculate the $\mathcal{L}_{invariant}(D_t^{new}, D_t^{buf}, D_t^{new,buf})$ loss, respectively. Both of them improve the performance of IFO further, but their combination (which is IFO++) achieves the best performance.

## A.8    ABLATION ANALYSIS OF HYPER-PARAMETERS IN IFO

Based on the results in In the experiments of Figure 4 and Figure 5. we set $s$ to 5 as it achieves the best performance with less compute. Also, we set the storage rate as 0.25 and $r_1$ as 0.75 and $r_2$ as 0.5.

## A.9    ANALYSIS OF THE INFLUENCE OF LEARNING INVARIANT FEATURES

We conduct experiments on the CIFAR100 dataset (10 tasks) and the buffer size of 2k. From Figure 6, we observe that the performance of the first two abilities of OCM and IFO are non-decreasing and bigger than 70 at the end. But the performance of the last ability is decreasing and is smaller than

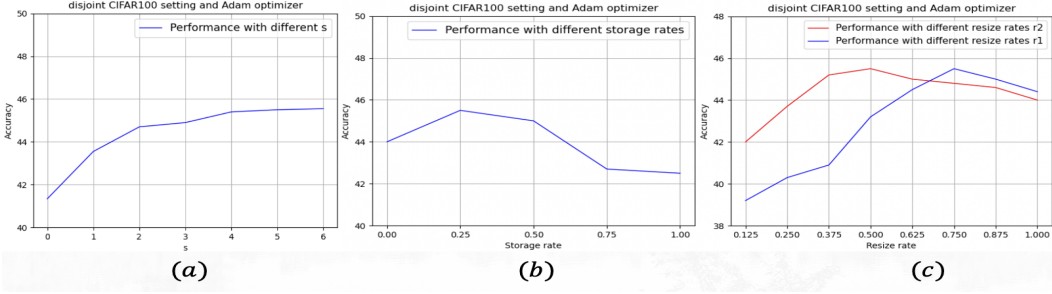

$(a)$ $\qquad\qquad\qquad\qquad (b)$ $\qquad\qquad\qquad\qquad (c)$

Figure 4: (a) results of CIFAR100 with different $s$. (b) results of CIFAR100 with different storage rate. (c) results of CIFAR100 with different resize rates $r_1$ and $r_2$. We set $B$ as 2k for all experiments.

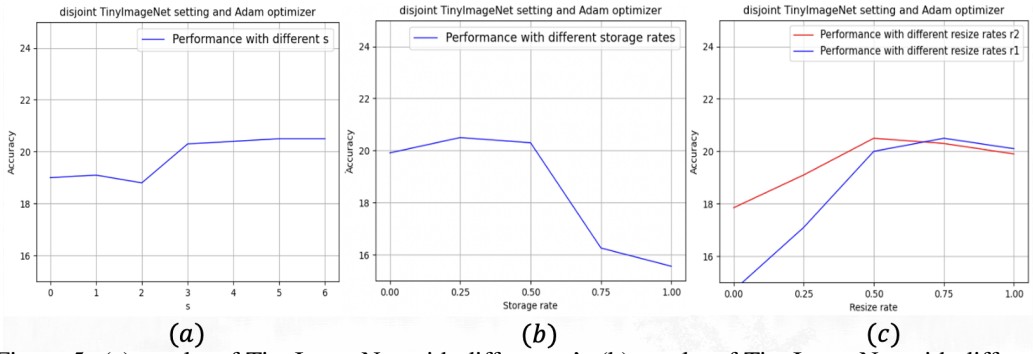

Figure 5: (a) results of TinyImageNet with different $s$'. (b) results of TinyImageNet with different storage rates. (c) results of TinyImageNet with different resize rates $r_1$ and $r_2$. We set $B$ as 2k for all experiments.

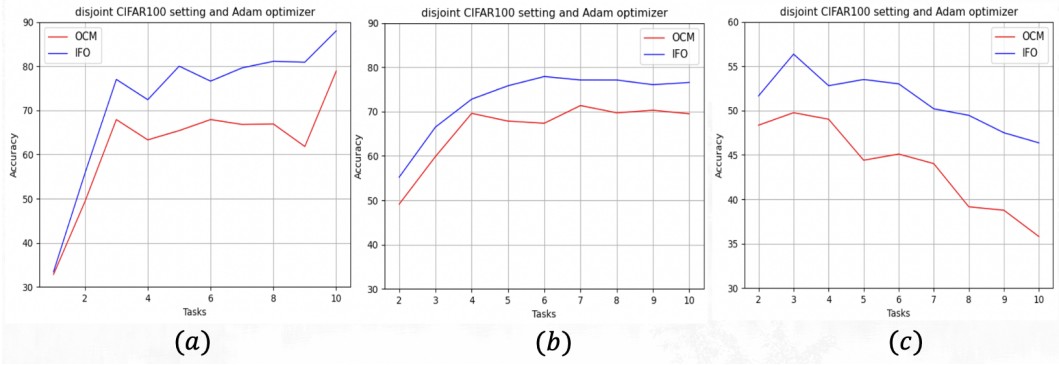

Figure 6: (a) the performance of learning new tasks. (b) the performance of maintaining learned decision boundaries. (c) the performance of establishing decision boundary between new classes and previous classes.

50 at the end (OCM (35.8) and IFO (46.36)). Also, we find that the average accuracy of OCM and IFO are 35.0 and 45.5 respectively in the Table 1. That means the main limitation of the performance of the current model is in establishing boundaries between classes from different tasks.

