# OpenReview forum: "Learning Invariant Features for Online Continual Learning"
_ICLR.cc/2023/Conference — Submitted to ICLR 2023_

### Official Review · Reviewer_jHRu · 2022-10-23

**Confidence:** 5
**Correctness:** 3
**Technical Novelty And Significance:** 4
**Empirical Novelty And Significance:** Not applicable
**Recommendation:** 8

**Clarity, Quality, Novelty And Reproducibility:**

The paper is well written and presents an original contribution to the field of online continual learning with strong experimental validation.
However, the paper lacks some clarity, especially in section 3, where some concepts are assumed known and not really explained (e.g. the memory buffer).
Also, it is not clear in what way the implemented augmentations relate to the more theoretical definition of invariance in section 3 (eq. 2).
In this regard, although the intuition of the two proposed augmentations seems clear, in practice, it is not convincing that this is the sort of invariance that may want to achieve. For example, randomly modifying the colour in the proposed way, may lead the model to just ignore colour in general and focus only on textural features.
However, colour undoubtedly contains important discriminative information. Do you have some evidence or further intuition on why this is sensible? (Also concerning the lambda sampled from a Beta distribution.)
Concerning the background augmentation (Aug_plus), this introduces some spurious noisy features around the "seam" or remove biases (related to the context or the scale) that are helpful for classification (or introduce new unrealistic and harmful biases).



**Strength And Weaknesses:**

Strenghts:
- Original approach with some theoretical justification and intuition
- Experimental results are convincing

Weaknesses:
- The two data augmentations are somewhat "handcrafted"
- Some parts are hard to follow (Section 3) without knowing OCM



**Summary Of The Paper:**

This paper presents a new approach for online continual learning extending the OCM method introduced by Guo et al. (2022). The proposed approach consists in an optimisation problem amd loss function that not only enforces to learn holistic representations of the data (as in OCM) but also class-invariant features. The idea is to simulate images of objects in different environments (via data augmentation), which alter the background of the images but also the colour of the objects themselves.
Experimental results show the effectiveness of the approach for different image classification benchmarks and two class-incremental learning protocols (disjoint and i-Blurry). The proposed method outperforms the state of the art in accuracy on these datasets.


**Summary Of The Review:**

Overall, the paper proposes an interesting and original contribution to continual online learning with rather thourough experimental validation on several difficult benchmarks outperforming the state of the art.

---

> ### Author Response · Authors · 2022-11-18
> **Response to Reviewer jHRu**
>
> Thank you for your comments. We have revised the paper in various places to make it easier to understand. The theoretical definition essentially says the model should learn features that are truly correlated with the class labels (e.g., the shape). We create augmented images with different colors and align them with the original image as we want the model to learn invariant features (e.g. shape) rather than simple features like color to form the representation. The latter is easier to happen as color feature is a so-called fast-to-learn feature. Lambda sampled from a Beta distribution ensures that we get augmented images with diverse colors. We know the color feature is important so the original image is included in the calculation of the invariant loss. That means the model can still learn some color features but mainly using a specific color of the object/background to form the representation is penalized. Also, in the background augmentation, we resize the original image and add a new background around it rather than cut and replace its original background with the new one. So the class object is protected and introducing some noisy/diverse backgrounds around it does not ruin it. Also, we propose a better sampling strategy for background augmentation in IFO++ (Sec.4.2, highlighted in blue).

---

### Official Review · Reviewer_K6JM · 2022-10-24

**Confidence:** 4
**Correctness:** 3
**Technical Novelty And Significance:** 2
**Empirical Novelty And Significance:** 3
**Recommendation:** 5

**Clarity, Quality, Novelty And Reproducibility:**

This paper is well written and the proposed method looks reproducible. The authors analyze the problems of online continual learning from a novel perspective. However, the novelty of this paper is incremental.

**Strength And Weaknesses:**

Strengths :

1. This paper analyze the problems of online continual learning from a novel perspective.
2. The method is easy to follow, and the performance is improved to a certain extent.

Weaknesses :

1. The novelty of this paper is incremental. Data augmentation methods have been proposed a lot in continual learning, such as [1][2]. The authors should compare the proposed data augmentation methods with these methods.
2. The idea of saving more samples proposed in the paper has already been proposed in continual learning, such as image compression [3], and the authors did not compare with the method.
[1] Class-Incremental Learning via Dual Augmentation, NeurIPS 2021
[2] Online Continual Learning Under Extreme Memory Constraints,ECCV2020
[3] Memory Replay with Data Compression for Continual Learning, ICLR 2022


**Summary Of The Paper:**

In this work, the authors argue that learning only holistic representations is still insufficient. The learned representations should also be invariant and those features that are present in the data but are irrelevant to the class (e.g., the background information) should be ignored for better generalization across tasks. This new condition further boosts the performance significantly. This paper proposes several strategies and a loss to learn holistic and invariant representations and evaluates their effectiveness in online CL.

**Summary Of The Review:**

The paper is well written and the authors analyze the problems of online continual learning from a novel perspective. However, the proposed method is not novel enough.

---

### Official Review · Reviewer_SrHu · 2022-10-25

**Confidence:** 4
**Correctness:** 3
**Technical Novelty And Significance:** 2
**Empirical Novelty And Significance:** 3
**Recommendation:** 6

**Clarity, Quality, Novelty And Reproducibility:**

This paper is mostly clear and can be followed with effort. Results look as though they could be reproduced if the code and datasets are made public, as the authors have promised to do upon publication. The level of novelty is low as the contributions are modest refinements to the existing OCM method.

The experimental results section can be improved by increasing the uniformity of methods compared. For instance, there are different baselines across the disjoint and blurry settings, it's not clear why the same baselines weren't used across all.

Minor points regarding clarity:
1) Page 7 - what is the "ER" baseline? It is not described as far as I could see
2) Page 8 - in figure 2(a), what does "CR" stand for? Is it another name for the "IFO" method?
3) Page 8 - the eigenvalues of IFO and OCM are claimed to be very different even though they look similar in the plot of appendix 4.


**Strength And Weaknesses:**

The strength of this work is that IFO's empirical results show improved accuracy and decreased forgetting on the 4 image datasets tested on. And stable/time-invariant representation learning is an important problem in both continual learning and in information retrieval where representational shifts necessitate reindexing large datasets.

The weakness of this work is that makes several smaller contributions -- combining the loss function of OCM with a correlation-based loss to induce embedding stability, and adding a novel image augmentation scheme to create synthetic backgrounds -- rather than making one large contribution.


**Summary Of The Paper:**


This paper proposes a fix for the catastrophic forgetting problem (CF) in continual learning (CL) with neural networks. The authors propose an extension of OCM (Guo, 2022), a method that seeks to learn features which preserve mutual information with the original inputs, the outputs, and the previously learned representations over the course of continual learning.

The extension proposed in this paper is called "IFO" which stands for Invariant Feature learning for Online CL. IFO is presented as an add-on to OCM that aims to learn "invariant" features which are defined as being both discriminative and present across different many different environments. To do this, IFO incorporates additional image augmentations while training under the OCM loss. Specifically, more color augmentations are used, and an augmentation which superimposes backgrounds from other images is used to create a more diverse set of contexts for objects being classified.

The paper evaluates IFO within 2 continual learning paradigms: 1) task disjoint, and 2) incremental-blurry task continual learning.
In the task disjoint setting, IFO has increased accuracy over other the CL methods on 4 datasets image datasets -- MNIST, CIFAR10/100, TinyImageNet. IFO outperforms all other CL baselines in terms of accuracy, including OCM which it shares a loss function with. IFO also has a lower forgetting rate.  In the incremental-blurry setting, IFO modestly improves over the baseline method, CLIB.

Finally, the paper presents some ablation experiments to determine which aspects of IFO improve performance most. In these experiments, other image augmentation strategies like CutMix and MixUp are used in place of the color and background augmentations used by IFO.


**Summary Of The Review:**

The empirical results for IFO look promising, but the methodological novelty and theoretical motivations are not big enough to advocate for strong acceptance to ICLR.  However, I am weakly in favor of acceptance to ICLR.

---

### Official Review · Reviewer_YuYN · 2022-10-26

**Confidence:** 3
**Correctness:** 3
**Technical Novelty And Significance:** 3
**Empirical Novelty And Significance:** 3
**Recommendation:** 8

**Clarity, Quality, Novelty And Reproducibility:**

**Awkward/incorrect parts**

While the writing is clear overall, there are occasional grammatical/structural problems that require careful editing. Example:

Sec 5.2: E.g., “Here we do not use the full IFO as our main contribution is the $L_{invariant}$
loss and IFO has all components of OCM.”

Also, shouldn't the main loss be:

$ L_{all} = L_{holistic} + L_{invariant} $

instead of:

$ L_{holistic} = L_{all} + L_{invariant}? $

**Novelty**

Invariance objectives have been explored for offline scenarios e.g., for distributional shifts, but application of invariance objective for continual learning seems novel.


**Strength And Weaknesses:**

**Strengths**

[S1] The proposed method of learning invariant features is important for continual learning. Improving invariance by constraining augmented views (environments) to have similar representations is sensible and empirically shown to improve generalization.

[S2] The experiments clearly show better generalization to new tasks as well as lower forgetting of older tasks, all stemming from the proposed invariant feature learning. With clear benefits on both disjoint and blurry tasks, this technique should be the baseline to beat for newer works.

[S3] The method shows large accuracy gains on multiple buffer sizes, showing that the invariance objective has better sample efficiency too.

**Weaknesses**

[W1] Since the choice of the augmentations (Aug color, Aug plus) are not clearly explained at the beginning, they seem arbitrary. Only in Sec 5.3, other schemes are discussed. I think the rationality for augmentation choices should be explained in Sec 4.1.

[W2] The claim: ‘invariance is not important for single/multitask learning due to i.i.d. assumption, is not entirely true. Single/multitask learning can have ood assumptions too, and there are multiple works already addressing invariant representation learning albeit in a non-continual learning setting e.g., IRM [1]

[W3] Invariance is probably better tested when there is a distributional shift. The experiments do not tackle sub-population or domain shifts, it only tackles shift in terms of new classes. But realistically, continual data streams even from same class may change over time (e.g., due to seasonal changes, inclusion of more domains e.g., race, noise characteristics etc), but this is not studied in this work.

[1] Arjovsky, Martin, et al. "Invariant risk minimization." arXiv preprint arXiv:1907.02893 (2019).

**Summary Of The Paper:**

The paper argues for invariant representations to enable continual learning. It proposes representations that generalize across environments to achieve invariance and shows that this helps reduce catastrophic forgetting.


**Summary Of The Review:**

The paper clearly shows that adding the invariance term helps generalization to new classes while showing lower forgetting. This is an important contribution, so I am leaning towards acceptance. The main weakness would be not studying sub-population/domain shifts, however, I do not think this is critical for acceptance. Also, the paper needs to be carefully edited for clear acceptance.

---

> ### Author Response · Authors · 2022-11-21
> **Response to Reviewer YuYN (1/2)**
>
> Response to [W1]: Thanks for the suggestion. We actually explained the rationales for the two augmentations in Section 4.1, e.g., our aim is to create augmented images that “have the same class semantics as the original input but diverse variant features” It is true that Section 5.3 has more interesting discussions in comparison with some other augmentations. We have revised Section 4.1 and referred readers to more discussions on the topic in Section 5.4 (highlighted in blue).
>
> Response to [W2]: Yes, we agree if OOD or data shift is considered, then invariance is also important in traditional supervised learning. We have revised the paper by including this and the citation [1] in footnote 2.
>
> Response to Awkward/incorrect parts: We have read the paper again and fixed the minor issues.

---

### Decision · Program_Chairs · 2023-01-20

**Decision:**

Reject

**Justification For Why Not Higher Score:**

The main problem with the paper is the lack of clear novelty with respect to prior work in offline and online incremental learning, as well as the fact that the proposed data augmentations are offered with little-to-no justification for *why* they should encourage the type of invariance needed for online CL. These problems, combined with more minor problems with clarity, justify the reject score.

**Justification For Why Not Lower Score:**

N/A

**Metareview: Summary, Strengths And Weaknesses:**

# Summary of Contribution

This paper describes an approach to online continual learning based on learning of invariant features. The authors propose two data augmentations combined with an optimization objective in order to encourage learning such invariant features. Augmented image features are aligned with the original images in order to measure class invariance in the invariance loss. The authors identify Local Sampling Bias as a cause for overfitting (i.e. learning variant features from incoming batches) and propose to correct this via targeted exemplar sampling. Experimental results are performed on two online class-incremental learning scenarios on MNIST, CIFAR-10, CIFAR-100, and TinyImageNet.

# Strengths

+ **Importance of Invariant Features**: The empirical results in the paper indicate the importance of invariant features. The authors consider two different scenarios: the standard disjoint-class one, and another "blurry" one in which classes can be repeated. The proposed approach outperforms compared approaches on selected benchmarks.

+ **Sample Efficiency**: The experimental results further demonstrate the effectiveness over a range of exemplar memory sizes, which distinguishes it from the compared approaches.

# Weaknesses

+ **Clarity**: The overall narrative of the paper is reasonably clear, but there are grammatical and typographical errors throughout. Some concepts in section 3 are assumed known/clear (e.g. "memory buffer") but should be clearly defined. The theoretical motivations and justifications for the specific augmentations are unclear (i.e. why should these two and their combination lead to the type of invariance needed for online CIL?). The paper is in need of revision for readability.

+ **Marginal Contributions**: The main contribution of the work is largely supported by two data augmentations, one of which is an color augmentation that is offered without much justification, and the other is a center-patch replacement augmentation hand-crafted to force the model to concentrate less on background features. The use of data augmentations (via a parallel network branch that slowly learns invariant features via self-supervised learning) has been explored in the online scenario in DualNet:

       Pham Q et al. Dualnet: Continual learning, fast and slow. Advances in Neural Information Processing Systems, 2021.

and the use of rotations to learn task-invariant features in the offline CIL case in PASS:

    Zhu F et al. Prototype augmentation and self-supervision for incremental learning. CVPR, 2021.

and neither of these works is discussed nor compared with (in the DualNet case especially) in the submitted paper. Moreover, the exemplar retrieval strategy proposed to compensate sampling bias is quite similar to the prototype selection used in the offline case in SSRE:

    Zhu K et al. Self-Sustaining Representation Expansion for Non-Exemplar Class-Incremental Learning. CVPR, 2022.

In brief, many of the contributions in the submitted manuscript are incremental variants of ideas that have already been circulating in the continual learning community.

+ **Heuristic Nature of Augmentations**: The two data augmentations ($Aug_{color}$ and $Aug_{plus}$) are somewhat ad hoc and are never clearly justified or motivated. The idea behind $Aug_{plus}$ is clear: inter-class center-crop replacement prevents models from attending to background features which might otherwise be incidentally discriminative for a specific class, but are not task-invariant. This is a somewhat brute-force approach, however, which might be exploiting the highly center-biased nature of all of the datasets considered in the experimental evaluation (CIFAR and TinyImageNet images have a strong bias towards centered objects). Similar to the last point, there is some previous work on using saliency (and thus marginalizing background influence) to guide feature learning going back to LwM:

      Dhar P, Singh RV, Peng KC, Wu Z, Chellappa R. Learning without memorizing. CVPR, 2019.

and more recently in RRR:

    Ebrahimi S, Petryk S, Gokul A, Gan W, Gonzalez JE, Rohrbach M, Darrell T. Remembering for the right reasons: Explanations reduce catastrophic forgetting. ICLR, 2021.

# Summary

The contributions of the proposed approach have limited novelty. The data augmentations are not well motivated, except for the $Aug_{plus}$ which is handcrafted to encourage learning object over background features. Moreover, the authors do not do an adequate job of contextualizing the novelty of their approach with respect to prior work on saliency-guided and data augmentation-based approaches to learning invariant features (see above). These issues, along with the problems of clarity pointed to by reviewers, make it fall short of the bar for acceptance at ICLR due to the limited novelty of the proposed approach. The key contributions of the paper are incremental variations of ideas circulating already for some time in the CIL community.